# Correlation of Yes-Associated Protein 1 with Stroma Type and Tumor Stiffness in Hormone-Receptor Positive Breast Cancer

**DOI:** 10.3390/cancers14204971

**Published:** 2022-10-11

**Authors:** Yangkyu Lee, Soong June Bae, Na Lae Eun, Sung Gwe Ahn, Joon Jeong, Yoon Jin Cha

**Affiliations:** 1Department of Pathology, Gangnam Severance Hospital, Yonsei University College of Medicine, Seoul 06273, Korea; 2Institute of Breast Cancer Precision Medicine, Yonsei University College of Medicine, Seoul 06273, Korea; 3Department of Surgery, Gangnam Severance Hospital, Yonsei University College of Medicine, Seoul 06273, Korea; 4Department of Radiology, Gangnam Severance Hospital, Yonsei University College of Medicine, Seoul 06273, Korea

**Keywords:** breast neoplasms, elastography, immunohistochemistry, pathology, YAP1 (yes-associated protein 1)

## Abstract

**Simple Summary:**

YAP1 is an oncogene that can be activated by matrix stiffness, as it can act as a mechanotransducer. So far, only in vitro studies regarding YAP1 activation and matrix stiffness are present. We confirmed the activation of YAP1 in breast cancer using human breast cancer tissue and immunohistochemistry. Tumor stiffness was quantified by shear-wave elastography. Nuclear localization of YAP1 showed correlation with tumor stiffness in hormone-receptor positive (HR+) breast cancer. Also, tumors with non-collagen-type stroma showed an association between YAP1 expression and tumor stiffness. YAP1 expression, along with tumor stiffness, may serve as a prognostic candidate in HR+ breast cancer.

**Abstract:**

(1) Background: Yes-associated protein 1 (YAP1) is an oncogene activated under the dysregulated Hippo pathway. YAP1 is also a mechanotransducer that is activated by matrix stiffness. So far, there are no in vivo studies on YAP1 expression related to stiffness. We aimed to investigate the association between YAP1 activation and tumor stiffness in human breast cancer samples, using immunohistochemistry and shear-wave elastography (SWE). (2) Methods: We included 488 patients with treatment-naïve breast cancer. Tumor stiffness was measured and the mean, maximal, and minimal elasticity values and elasticity ratios were recorded. Nuclear YAP1 expression was evaluated by immunohistochemistry and tumor-infiltrating lymphocytes (TILs); tumor-stroma ratio (TSR) and stroma type of tumors were also evaluated. (3) Results: Tumor stiffness was higher in tumors with YAP1 positivity, low TILs, and high TSR and was correlated with nuclear YAP1 expression; this correlation was observed in hormone receptor positive (HR+) tumors, as well as in tumors with non-collagen-type stroma. (4) Conclusions: We confirmed the correlation between nuclear YAP1 expression and tumor stiffness, and nuclear YAP1 expression was deemed a prognostic candidate in HR+ tumors combined with SWE-measured tumor stiffness.

## 1. Introduction

Yes-associated protein-1 (YAP1) is a transcription coactivator and downstream effector of the Hippo pathway, along with the PDZ-binding motif (TAZ) [1]. Normally, YAP/TAZ activity is inhibited by phosphorylation, resulting in cytoplasmic retention and subsequent sequestration of YAP/TAZ [2,3]. However, when the Hippo pathway is disrupted, YAP/TAZ moves into the nucleus of the cells, interacts with the TEA domain transcription factor (TEAD), and induces the activation of multiple genes involved in cell proliferation, apoptosis, and survival [2]. Overexpression of YAP1 correlates with poor prognosis in human cancers, including ovarian cancer [4,5], non-small cell lung cancer [6], esophageal cancer [7], and breast cancer [8,9]. Independent of Hippo pathway dysregulation, YAP1 can be activated via mechanical stress, such as stiffness of the tumor microenvironment and/or tissue adhesion [10]. Activated YAP1 enhances the proliferation, survival, metastasis, and drug resistance of cancer cells [11].

In breast cancer, YAP1 overexpression in tumor promotion and patient prognosis have been controversial. Studies have reported that YAP1 functions as a tumor suppressor in breast cancer [12,13] and is associated with favorable outcomes [14], whereas others have suggested that YAP1 acts as an oncogene [15,16]. A previous study analyzed the nuclear expression of YAP1 in triple-negative breast cancer (TNBC) and showed that adverse prognosis was associated with nuclear YAP1 expression [8].

Shear-wave elastography (SWE) is an ultrasound (US) technique that can quantitatively measure the stiffness of a target lesion in kilopascals (kPa) in vivo [17]. SWE is now widely used in patients with breast masses for the differentiation of benign and malignant lesions [18]. Malignant breast lesions usually have higher elasticity values than benign breast lesions, and invasive carcinoma has an even higher elasticity value than carcinoma in situ [18]. Among several quantitative SWE measurements (mean, maximal, and minimal), maximal elasticity is known to be well correlated with lesion stiffness [19].

Taken together, we speculated that quantitatively measured tumor stiffness by SWE might correlate with YAP1 activation that could be recognized by immunohistochemistry (IHC) of the nuclear expression of YAP1. To date, in vivo studies on YAP1 activation and matrix stiffness in human tissue samples have not been performed. In this study, we analyzed the correlation between nuclear YAP1 expression and tumor stiffness in breast cancer.

## 2. Materials and Methods

### 2.1. Patients

We retrospectively collected tumor tissues from patients who underwent upfront curative surgery followed by adjuvant treatments for breast cancer at Gangnam Severance Hospital in Seoul, Korea, from January 2012 to April 2020. Clinical and pathological data were obtained by reviewing the electronic medical records. Stages were determined according to the 8th edition of the American Joint Committee on Cancer staging system.

The inclusion and exclusion criteria were as follows:Inclusion criteria:
Patients aged ≥ 20 yearsInvasive breast cancer confirmed by pathological diagnosisPreoperative elastography resultsAvailable YAP1 IHC staining of the resected tissueExclusion criteria:
Any other carcinoma in situOther cancer history (except for thyroid cancer and carcinoma in situ)Inaccessible electronic medical recordsReceived Neoadjuvant Chemotherapy (NAC)

The following data were collected: age at surgery, breast and axillary lymph node surgery, adjuvant treatments, tumor size, lymph node status, histological grade (HG), estrogen receptor (ER) status, progesterone receptor (PR) status, human epidermal growth factor receptor 2 (HER2) status, lymphovascular invasion (LVI), Ki67 leveling index, and tumor-infiltrating lymphocytes (TILs).

IHC staining was performed using light microscopy (BX53 upright microscope; Olympus, Tokyo, Japan). Nuclear staining values of 1% or higher were considered positive for ER (clone 6F11; dilution 1:200; Leica Biosystems, Wetzlar, Germany) and PR (clone 16; dilution 1:500; Leica Biosystems) [20]. HER2 (clone 4B5; dilution 1:5; Ventana Medical System, Oro Valley, AZ, USA) staining was performed according to the guidelines of the 2018 American Society of Clinical Oncology/College of American Pathologists [21]. Only samples with strong and circumferential membranous HER2 immunoreactivity (3+) were considered positive, whereas those with 0 and 1+ HER2 staining were considered negative. Cases with equivocal HER2 expression (2+) were further evaluated for HER2 gene amplification by silver in situ hybridization (SISH). Positive nuclear Ki-67 (clone MIB; dilution 1:1000; Abcam, Cambridge, UK) staining was assessed based on the percentage of positive tumor cells, defined as the Ki-67 labelling index.

The specimens were categorized into the following subtypes:Hormone receptor-positive (HR+) HER2 negative (HER2−): ER and/or PR positive and HER2 negativeHR+HER2+: ER and/or PR positive and HER2 overexpressed and/or amplifiedHER2: ER and PR negative and HER2 overexpressed and/or amplifiedTriple-negative breast cancer (TNBC): ER, PR, and HER2 negative

The TIL levels were concurrently evaluated according to the guidelines suggested by the International TIL Working Group [22]. Except for polymorphonuclear leukocytes, other mononuclear cells, including lymphocytes and plasma cells, were counted. For statistical analysis, a 30% cutoff was applied to divide patients into low TIL (<30%) and high TIL (≥30%) groups [23].

### 2.2. Shear Wave Elastography

Breast US examinations were performed using the Aixplorer US system (SuperSonic Imagine, Aix-en-Provence, France), which was equipped with a 4–15-MHz linear-array transducer by one of four radiologists with 5–10 years of breast US examination experience. The investigators were aware of the clinical examination and mammography results at the time of examination. After obtaining gray-scale US images, SWE images were obtained for breast masses that were scheduled for biopsy or surgical excision. The built-in region of interest (ROI) (Q-box; SuperSonic Imagine) of the system was set to include the lesion and surrounding normal tissue, which was displayed as a grayscale image overlaid with a semitransparent color map of tissue stiffness ranging from dark blue (lowest stiffness) to red (highest stiffness), from 0–180 kPa. Areas of black on the SWE images represented tissue in which no shear waves were detected. Fixed 2 × 2-mm ROIs were placed by an investigator over the stiffest part of the lesion, including the immediate adjacent stiff tissue or halo. The system calculated the mean elasticity value in kPa for the mass. Tumor stiffness was measured in terms of mean, maximal, and minimal elasticity values, and elasticity ratio, which was comparable to that of the adjacent fat tissue (Figure 1).

### 2.3. Evaluation of Tumor–Stroma Ratio and Tumor Stroma Subtyping

Hematoxylin and eosin (H&E) stained slides from formalin-fixed paraffin tissue (FFPE) were used to evaluate the tumor–stroma ratio (TSR). All H&E-stained slides from each case were reviewed for TSR evaluation. TSR is defined as tumor cellularity relative to the surrounding stroma in the overall tumor bed [24]. The TSR assessment was conducted using scoring percentages in 10% increments (10%, 20%, 30%, etc.). For statistical analysis, cases with <50% TSR were assigned to the low TSR group and those with ≥50% TSR were assigned to the high TSR group. The tumor stroma was further categorized into two subgroups based on the collagenous stromal component as follows (Figure 2):Collagen: dense collagenous fibrosis without cellular componentsNon-collagen: non-collagenous stroma, including mesenchymal, inflammatory, or mucinous components

### 2.4. YAP1 IHC and Interpretation

H&E slides from resected breast cancer specimens were reviewed, and one representative whole slide was selected for further YAP1 IHC. Briefly, 3-µm thick tissue sections were cut from the FFPE tissue block of the most representative section. After deparaffinization and rehydration with graded xylene and alcohol solutions, IHC was performed using a Ventana Discovery XT Automated Slide Stainer (Ventana Medical System, Tucson, AZ, USA). Cell conditioning 1 (CC1) buffer (citrate buffer, pH 6.0; Ventana Medical System) was used for antigen retrieval. Appropriate positive and negative controls were included in this study. Whole tissue slides were stained with an anti-YAP1 antibody (clone 63.7; dilution 1:200; Santa Cruz Biotechnology, Dallas, TX, USA). After staining, nuclear YAP1 expression was assessed by two breast pathologists (YL and YJC; 400× magnification). YAP1 expression was evaluated in the nuclei of the tumor cells. Nuclear staining was evaluated by the H-score, which was obtained by multiplying the staining intensity (0, 1, 2, or 3) by the percentage of stained area (%). The intensity of nuclear staining of myoepithelial cells was assigned a value of 2+ and used as an internal control. Weaker and stronger signals were assigned the values of 1+ and 3+, respectively. (Figure 3). For further analysis, an H-score of 20 was applied as the cutoff to determine the YAP1-negative and YAP1-positive groups. IHC results were interpreted blindly, without any information regarding clinical parameters or outcomes.

### 2.5. Statistical Analysis

Data were analyzed using SPSS (version 23.0; SPSS Inc., Chicago, IL, USA). Statistical significance was set at *p* < 0.05. Continuous variables between the two groups were compared using the Student’s *t*-test or Mann–Whitney U test. Categorical variables were compared using the chi-square or Fisher’s exact tests. Univariate and multivariate analyses with linear and logistic regression were performed to assess the correlation between YAP1 expression and elasticity.

## 3. Results

### 3.1. Study Population

The baseline clinicopathological characteristics of the patients are shown in Table 1. Among the 488 patients, the median age was 52 years (range, 24–91 years), and 429 (85.9%) were diagnosed with invasive ductal carcinoma. The median follow-up period was 19.1 months (interquartile range [IQR], 12.1–30.2 months). All patients received upfront surgery followed by adjuvant treatment. Further information regarding breast surgery and adjuvant treatments is shown in Table A1. Most patients received the breast-conserving surgery (*n* = 363, 74.4%), sentinel lymph node biopsy (*n* = 429, 87.9%), hormone therapy (*n* = 400, 81.9%), and radiotherapy (*n* = 397, 79.7%). In addition, 51% of patients (*n* = 249) received anthracycline- or taxane-based chemotherapy. No patients died. Two cases of local recurrence occurred (0.4%). The mean tumor size was 1.9 cm and 307 patients were assigned to the pT1 stage. The HR + HER2− subtype was the most predominant (*n* = 363, 74.4%), followed by TNBC (*n* = 46, 9.4%), HR + HER2+ (*n* = 45, 9.2%), and HER2 (*n* = 34, 7.0%). Lymph node metastasis was observed in 107 (21.9%) patients. Most tumors were histological grade II (*n* = 305, 62.5%). Collagenous stroma was found in 197 (40.4%) tumors and the median TSR was 70 (IQR 40–80).

### 3.2. Different Elasticity of Breast Cancer According to Pathologic Parameters

We analyzed the differences in elasticity according to histological subtype, molecular subtype, HR status, TIL level, stroma type, and YAP1 expression (Table 2). Among the histological subtypes, ILC showed a significantly lower elasticity ratio than the other subtypes (9.3 ± 5.8 in ILC vs. 13.3 ± 10.9 in IDC and 17.7 ± 18.0 in others; *p* = 0.027). YAP1-positive tumors were significantly stiffer, showing higher mean elasticity (166.9 ± 73.3 kPa vs. 149.1 ± 66.2 kPa; *p* = 0.013) and maximal elasticity (190.1 ± 77.5 kPa vs. 169.9 ± 74.1 kPa; *p* = 0.010) values than those of YAP1-negative tumors. A higher minimal elasticity value was observed in YAP1-positive tumors but was not statistically significant (141.5 ± 153.6 kPa vs. 115.7 ± 59.9 kPa; *p* = 0.075). The elasticity ratio did not differ between YAP1-positive and YAP1-negative tumors. The low TIL group showed consistent stiffness, revealed by significantly higher mean (163.6 ± 67.8 kPa vs. 121.2 ± 59.4 kPa; *p* < 0.001), maximal (131.6 ± 100.2 kPa vs. 91.5 ± 51.7 kPa; *p* < 0.001), and minimal (186.2 ± 74.5 kPa vs. 139.0 ± 66.2 kPa; *p* < 0.001) elasticity values and elasticity ratios (14.18 ± 12.2 vs. 10.9 ± 8.5; *p* = 0.002) than those of the high TIL group. Regarding TSR, tumors from the high TSR group were stiffer and had significantly higher mean (141.2 ± 69.9 kPa vs. 159.2 ± 67.1 kPa; *p* = 0.008), maximal (161.1 ± 77.2 kPa vs. 181.3 ± 73.8 kPa; *p* = 0.031), and minimal (108.0 ± 62.6 kPa vs. 128.3 ± 102.1 kPa; *p* = 0.008) elasticity values than tumors of the low TSR group.

There was no difference in stiffness or elasticity ratio between the molecular subtype, HR status, and stroma type.

### 3.3. Association between YAP1 Expression and Elasticity

On linear regression regarding the YAP1 H-score and stiffness in all patients, the mean and maximal elasticity values showed a significant positive correlation with the YAP1 H-score (Table A2). When YAP1 expression was analyzed as a binary parameter (positive/negative), the mean, maximal, and minimal elasticity values were correlated with YAP1 positivity in the univariate analysis. In the multivariate analysis, the maximal elasticity value was the only independent value that correlated with the YAP1 H-score as well as YAP1 positivity (Table A2).

### 3.4. Subgroup Analyses: YAP1 Expression and Tumor Stiffness Based on the HR Status and Stroma Type

Subgroup analysis was performed based on the HR status (Table A3). In HR+ tumors, YAP1 positivity was significantly correlated with tumor stiffness (mean, maximal, and minimal elasticity values); however, no correlation was found in HR- tumors. Multivariate analysis showed that the maximal elasticity value was independently correlated with YAP1 expression in HR+ tumors.

YAP1 expression and tumor stiffness were only correlated in the tumors of non-collagenous stroma (Table A4). The mean and maximal elasticity values were significantly correlated with YAP1 expression by univariate analysis, and the maximal elasticity value was independently correlated with YAP1 positivity.

## 4. Discussion

This is the first in vivo study to confirm that YAP1 activation is correlated with lesion stiffness, using a human cancer sample. In this study, nuclear YAP1 expression and tumor stiffness measured by SWE showed a significant correlation in linear and logistic regression analyses, which also means YAP1 expression could be a surrogate marker of tumor stiffness. Nuclear localization of YAP1 is known to play the role of an oncogene [2,16]. Controversies have been reported regarding the clinical significance of YAP1 expression in breast cancer [12,14,25,26]. However, in previous studies, the methodology for measuring YAP1 expression by IHC was not described in detail [12,14,25,26], and whether nuclear expression was identified could not be determined.

A previous study by our team showed that nuclear expression of YAP1 is a poor prognostic factor in patients with TNBC [8]. In that study, we used a tissue microarray (TMA) sample, which was one of the major limitations of that study because nuclear YAP1 expression was only focally present in the tumor; thus, YAP1 expression in the cancer tissue of a 3 mm-sized TMA core might not be representative. With limited TMA tissue, only the presence or absence of nuclear staining, regardless of positivity proportion, could have been evaluated, which divided patients into YAP1-low (negative or weak intensity) and YAP1-high (moderate or strong intensity) groups. In this study, we performed IHC on whole tumor slides and set the H-score to 20 as the YAP1 positivity cut-off.

The transcriptional regulator YAP1 acts as an oncogene in the dysregulated Hippo pathway, and YAP1 is also considered a mechanotransducer, which is activated by an increase in matrix stiffness [27]. Increased tissue stiffness is a strong risk factor for breast cancer progression. Previous studies using SWE in breast cancer showed that increased tumor stiffness was associated with poor prognostic factors, such as high histologic grade, large invasive size, LVI, and lymph node metastasis [28,29,30].

Normally, the stroma is composed of the extracellular matrix (ECM) and cellular components, including fibroblasts, endothelial cells, and immune cells. During cancer cell invasion of the stroma, the basement membrane between cancer cells and the stroma is disrupted and the stroma becomes fibrotic and activated; this phenomenon is called desmoplastic reaction [31]. Stromal fibroblasts in the TME are referred to as cancer-associated fibroblasts, and desmoplastic reactions make the ECM denser and stiffer, induced by connective fibers such as tenascin and fibronectin [32,33]. The tumor stroma plays a critical role in cancer cell growth, progression, metastasis, and drug resistance [31]. Stiffness of ECM also enhances drug resistance in breast cancer cells [34]. Qin et al. showed that stiffness-induced YAP1 nuclear translocation and mediated drug resistance gene expression, and also suggested that inhibition of YAP1 could regulate drug resistance of breast cancer cells [34].

The contribution of YAP1 in tumor immunity has not been clearly established. However, several studies have indicated that YAP1 expression is associated with increased immune cell infiltration in various solid tumors including pancreatic cancer [35], ovarian cancer [36], and hepatocellular carcinoma [37], etc. So far, direct correlation of matrix stiffness-YAP1- immunosuppression has not been well studied. Matrix stiffness itself contributes to immune evasion of tumor cells by reducing cellularity and function of cytotoxic T-cells, as well as enhancing immunosuppressive macrophage polarization [38]. A recent in vitro study showed that mechanical stiffness promotes dendritic cells and elicits an adaptive immune response in tumor immunity via TAZ, which is a YAP homologous [39].

An in vitro study showed that YAP1 was activated when breast cancer progressed from in situ carcinoma to invasive carcinoma [40]. This implies that disruption of the basement membrane and direct contact between tumor cells and stroma might play a role in YAP1 activation. In the present study, tumors with YAP1 positivity, low TILs, and high TSR showed significantly higher tumor stiffness than those with YAP1 negativity, high TILs, and low TSR. There was no difference in tumor stiffness based on the HR status, molecular subtype, or stromal type. A high TSR implies that the tumor cell itself comprises more than half of the tumor. TILs are stromal cells composed of inflammatory cells. In contrast to other mesenchymal components embedded in the stroma or tumor cells that are tightly attached to each other, TILs have no adhesive properties, which could explain the lower tumor stiffness in high-TIL tumors. In this study, most cases were invasive ductal carcinoma, of which the histologic subtype retained E-cadherin, which maintains cell–cell adhesion. Preserved cell–cell adhesion and high cellularity may contribute to increased tumor stiffness. Interestingly, invasive lobular carcinoma showed a lower elasticity ratio than other histologic variants. This could be explained by the loss of e-cadherin on tumor cells of invasive lobular carcinoma. Discohesive tumor cells might have loosened the cell–cell adhesion, and the difference in stiffness between the tumor and surrounding stroma might be smaller than that of invasive ductal carcinoma.

Regression analyses showed that YAP1 expression was significantly increased along with tumor stiffness, and YAP1 positivity showed a positive correlation with maximal elasticity value, which supports previous in vitro studies regarding YAP1 activation via matrix stiffness [11,41].

The correlation between YAP1 and stiffness in HR+ tumors is interesting. Usually, HR- tumors—HER2 and TNBC—bring substantial stromal TILs and commonly have a high histological grade. Abundant stromal TILs may lower tumor stiffness. As described above, lymphocytes do not adhere anywhere, leading to the loosening of cell–cell adhesion. Furthermore, in tumors with a high histological grade, tumor cells tend to have mesenchymal properties; otherwise, epithelial characteristics might be weakened [42]. However, considering that tumor stiffness did not differ among the molecular subtypes, a significant correlation between YAP1 expression and tumor stiffness in HR+ tumors might be a specific trait of HR+ tumors. Contrary to HER2 and TNBC tumors, which achieve a more frequent pathologic complete response following NAC, HR+ tumors are relatively poor responders to NAC [43,44]. Moreover, HR+ tumors have no specific predictive factors for target therapy—such as HER2 amplification for trastuzumab in HER2 tumors—and TILs have no predictive role in HR+ tumors in an NAC setting [45]. Increased tumor stiffness in breast cancer is known to be associated with poor prognostic factors [28,29]. Considering previous studies and our results together, we speculate that YAP1 could be a surrogate marker for prognosis or a therapeutic target in HR+ breast cancer, which should be validated by further research.

Further checks on the activated YAP1 and downstream signaling might have led to a more convincing conclusion to our study. However, it appears to be difficult validate YAP1 signaling effects accurately. Nuclear localization of YAP1 is known to activate multiple cancer-associated genes, and YAP1 activation with poor prognosis in various solid tumors is considered to be a result derived from all the complex effects of the cancer-associated gene that is put together. Moreover, as downstream cancer-associated genes are variable, selection bias might be inevitable.

At first, we hypothesized that tumors with collagenous stroma might be stiffer and show YAP1 positivity. However, the results were different from our expectations; there was no difference in tumor stiffness depending on the stroma type, and YAP1 positivity correlated with stiffness in tumors with non-collagenous stroma. Hence, we assumed that stromal cellularity, except TILs, does not affect tumor stiffness; rather, tumor cellularity appears to be more important for stiffness. A previous study using a 3D culture of breast cancer cells suggested that YAP1 and matrix stiffness is irrelevant under absence of stress fibers [40]. This finding is in line with our result that there was no correlation with YAP1 expression and stiffness in tumors with collagenous stroma, which had no relevant mesenchymal cellular components. Considering that YAP1 expression in non-collagenous stroma is correlated with tumor stiffness, YAP1 activation might be induced by mesenchymal cellular components, such as fibroblasts, along with matrix stiffness.

This study had two major limitations. First, the patient population was skewed with respect to histological and molecular subtypes—85.9% were invasive ductal carcinoma and 74.4% of patients were the HR + HER2− type. Also, the number of patients was not sufficient to make a solid conclusion. In the present study, we focused on the evaluation of YAP1 expression and tumor stiffness, which requires treatment-naïve whole tumor tissue for accurate and detailed microscopic evaluation. In cases of HER2+ or TNBC, NAC is widely performed in nearly half of the patients, even in small tumors, which explains the skewed HR+ tumor predominance, as most of patients with HR+ tumors receive upfront surgery. The insignificant correlation between YAP1 expression and HR- tumors might also originate from the relatively small number of HR- tumors and should be further investigated. Second, there were very few events. No patients died, and only two experienced recurrences. Owing to the lack of events, the clinical implications of YAP1 expression are limited. This could also be explained by the characteristics of the study cohort, where most cases were HR+ (*n* = 407, 83.4%), pT1 stage (*n* = 307, 62.9%), and pN0 (*n* = 381, *n* = 78.1%). In this study, the range of follow-up period was 0.3–43.3 months (median 19.1 months, IQR 12.1–30.2 months), which might not be long enough to evaluate the patients’ outcomes, considering HR+ tumors were the most predominant type. As previously reported, approximately 50% of patients with HR+ tumors experience late recurrence more than 5 years after the initial cancer diagnosis [46]. Further studies with a long follow-up period are required to validate the clinical implications of YAP1 expression in HR+ tumors. Authors should discuss the results and how they can be interpreted from the perspective of previous studies and of the working hypotheses. The findings and their implications should be discussed in the broadest context possible. Future research directions may also be highlighted.

## 5. Conclusions

In conclusion, we verified a positive correlation between YAP1 expression and tumor stiffness in breast cancer. This is the first in vivo study on human cancer samples. In particular, YAP1 expression and tumor stiffness were significantly correlated in HR+ tumors and in tumors with non-collagenous stroma. YAP1 expression may be a poor prognostic factor, but the clinical implications of YAP1 in HR+ tumors should be further validated. Furthermore, not only matrix stiffness and YAP1 activation, but also the effect of the interaction between stromal cellular components and YAP1 on YAP1 activation, should be investigated.

## Figures and Tables

**Figure 1 cancers-14-04971-f001:**
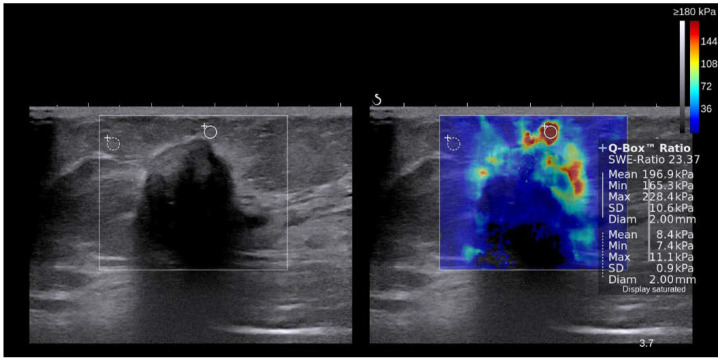
**Representative picture of shear wave elastography.** A 3.2 cm irregular hypoechoic mass was observed upon US examination. The overlaid semitransparent color reflects tissue stiffness, ranging from dark blue (lowest stiffness) to red (highest stiffness). In this tumor, tumor stiffness is present as mean (196.9 kPa), minimal (165.3 kPa), and maximal (228.4 kPa) elasticity values. The elasticity ratio was calculated by dividing the mean elasticity of the tumor (196.9 kPa) by that of the adjacent stroma (8.4 kPa).

**Figure 2 cancers-14-04971-f002:**
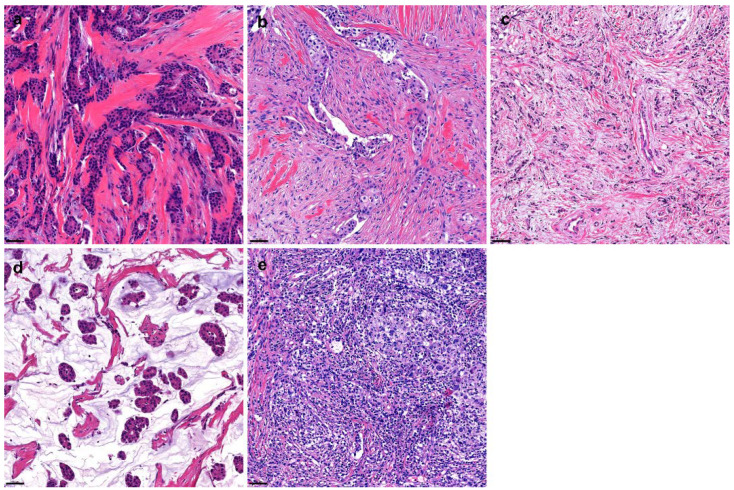
**Variable tumor stroma types of breast cancer.** The tumor stroma can be variably present. In this study, the stroma was divided into collagenous (**a**) and non-collagenous (**b**–**e**) types. The collagenous stroma is characterized by dense, pinkish, cellular collagenous fibrosis (**a**). Non-collagenous stroma includes stroma with mesenchymal cellular components (**b**), loose and sparse cellular components (**c**), mucinous components (**d**), or inflammatory components (**e**).

**Figure 3 cancers-14-04971-f003:**
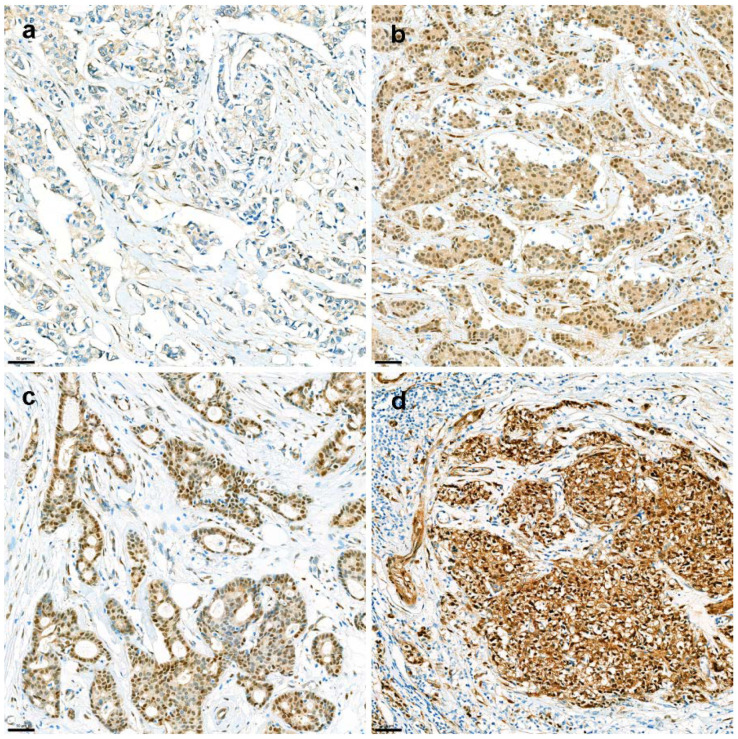
**Immunohistochemistry for nuclear YAP1 expression.** Nuclear YAP1 expression was evaluated in high-power fields (400× magnification) by two breast cancer pathologists. Expression intensity was graded as negative (**a**), weak (**b**), moderate (**c**), or strong (**d**).

**Table 1 cancers-14-04971-t001:** Basal characteristics of patients.

Variables	Patients (*n* = 488)
Age (years, median, range)	52 (24–91)
Histologic type	
Invasive ductal carcinoma	429 (85.9)
Invasive lobular carcinoma	21 (4.3)
Others	38 (7.8)
Tumor size (cm, mean ± SD)	1.9 ± 1.2
pT stage, %	
T1	307 (62.9)
T2-3	181 (37.1)
Molecular subtypes	
HR + HER2−	363 (74.4)
HR + HER2+	45 (9.2)
HER2	34 (7.0)
TNBC	46 (9.4)
LN metastasis	
Absent	381 (78.1)
Present	107 (21.9)
Histological grade	
I	101 (20.7)
II	305 (62.5)
III	82 (16.8)
TILs (%, mean ± SD)	19.3 ± 23.1
Tumor-stroma ratio (%, median, IQR)	70.0 (40.0–80.0)
Stroma type	
Collagen	197 (40.4)
Non-collagen	291 (59.6)
Ki67 labeling index (%, mean ± SD)	18.8 ± 21.9
Elasticity value (kPa, mean ± SD)	
Mean	155.0± 69.4
Minimal	122.9± 95.2
Maximal	176.1± 76.3
Ratio	13.5 ± 11.6
FU (months, median, IQR)	19.1 (12.1–30.2)
Death	0 (0.0)
Recurrence	2 (0.4)

SD, standard deviation; HR+, hormone receptor positive; HER2−, human epidermal growth factor receptor 2 negative; HER2−, HER2 negative; TNBC, triple-negative breast cancer; LN, lymph node; TILs, tumor-infiltrating lymphocytes; IQR, interquartile range; kPa, kilopascals; FU, follow-up period.

**Table 2 cancers-14-04971-t002:** Difference in tumor stiffness and elasticity ratio based on the clinicopathological parameters.

	Elasticity Values (kPa, Mean ± SD)						Elasticity Ratio	*p*
	Mean	*p*	Maximal	*p*	Minimal	*p*		
**Histologic subtype**								
IDC (*n* = 429)	153.5 ± 68.6	0.525	175.1 ± 75.5	0.530	122.3 ± 96.7	0.943	13.3 ± 10.9	0.027
ILC (*n* = 21)	145.8 ± 67.7		165.0 ± 74.3		119.1 ± 63.6		9.3 ± 5.8	
Other † (*n* = 38)	164.9 ± 65.7		186.7 ± 74.8		127 ± 60		17.7 ± 18.0	
**Molecular subtype**								
HR + HER2− (*n* = 363)	153.2 ± 66.8	0.128	175.4 ± 74.4	0.919	123.2 ± 100.8	0.973	12.7 ± 10.2	0.899
HR + HER2+ (*n* = 45)	153.9 ± 79.0		174.3 ± 83.3		112.8 ± 75.6		16.5 ± 17.9	
HER2 (*n* = 34)	161.8 ± 75.3		181.1 ± 81.4		126.2 ± 62.2		14.9 ± 11.6	
TNBC (*n* = 46)	155.1 ± 65.4		173.36 ± 71.6		124.9 ± 63.2		15.0 ± 13.1	
**HR status**								
HR- (*n* = 81)	159.7 ± 70.8	0.417	178.3 ± 76.3	0.716	127.6 ± 65.0	0.593	15.0 ± 12.3	0.200
HR+ (*n* = 407)	153.0 ± 67.8		175.0 ± 75.2		121.5 ± 97.8		13.1 ± 11.3	
**Stroma type**								
Collagenous (*n* = 197)	156.9 ± 68.1	0.457	180.0 ± 76.3	0.281	120.6 ± 64.3	0.705	12.9 ± 10.8	0.403
Non-collagenous (*n* = 191)	152.2 ± 68.4		172.5 ± 74.6		123.9 ± 108.2		13.8 ± 12.0	
**YAP1 positivity**								
YAP1-positive (*n* = 119)	166.9 ± 73.3	0.013	190.1 ± 77.5	0.010	141.5 ± 153.6	0.075	14.1 ± 10.3	0.470
YAP1-negative (*n* = 357)	149.1 ± 66.2		169.9 ± 74.1		115.7 ± 59.9		13.2 ± 12.0	
**TIL level**								
Low-TIL (*n* = 378)	163.6 ± 67.8	<0.001	131.6 ± 100.2	<0.001	186.2 ± 74.5	<0.001	14.18 ± 12.2	0.002
High-TIL (*n* = 110)	121.2 ± 59.4		91.5 ± 51.7		139.0 ± 66.2		10.9 ± 8.5	
**TSR**								
Low-TSR (*n* = 139)	141.2 ± 69.9	0.008	161.1 ± 77.2	0.031	108.0 ± 62.6	0.008	12.7 ± 10.8	0.364
High-TSR (*n* = 349)	159.2 ± 67.1		181.3 ± 73.8		128.3 ± 102.1		13.8 ± 11.8	

kPa, kilopascal; SD, standard deviation; IDC, invasive ductal carcinoma; ILC, invasive lobular carcinoma; HR+, hormone receptor-positive; HER2−, human epidermal growth factor receptor 2-negative; HER2−, HER2-negative; TNBC, triple-negative breast cancer; TIL, tumor-infiltrating lymphocyte; TSR, tumor-stroma ratio † 19 mucinous carcinomas, 7 tubular carcinomas, 6 papillary carcinomas, 2 cribriform carcinomas, 2 metaplastic carcinomas, 2 medullary carcinomas.

## Data Availability

All data generated or analyzed during this study are included in this research article.

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
