# Peer review of "Correlation of Yes-Associated Protein 1 with Stroma Type and Tumor Stiffness in Hormone-Receptor Positive Breast Cancer"

_cancers, 2022, doi:10.3390/cancers14204971_

Round 1

Reviewer 1 Report

This study aims at identifying whether a link could be found between the nuclear localisation of the oncogene Yes-associated protein 1 (YAP1) and the stiffness of breast tumors, expressing hormonal receptors or not. Nuclear localisation of YAP1 is examined by classical IHC methods on patients biopsies, and tissue stiffness is directly assessed on patients breast by shear wave elastography.

This is the main and only originality of this study, which is on the whole confirmatory of previously reported data, and which merely establishes a correlative link between both parameters. The number of patients enrolled in the study is too low to conclude that YAP1 could be a reliable cancer biomarker.

Author Response

Reviewer 1

This study aims at identifying whether a link could be found between the nuclear localisation of the oncogene Yes-associated protein 1 (YAP1) and the stiffness of breast tumors, expressing hormonal receptors or not. Nuclear localisation of YAP1 is examined by classical IHC methods on patients biopsies, and tissue stiffness is directly assessed on patients breast by shear wave elastography.

This is the main and only originality of this study, which is on the whole confirmatory of previously reported data, and which merely establishes a correlative link between both parameters. The number of patients enrolled in the study is too low to conclude that YAP1 could be a reliable cancer biomarker.

[Answer]

             Thank you for your comment. We also recognize the limitation of our study that number of patients relatively small to make a solid conclusion, and case number of each subtype is skewed toward hormone positive breast cancer. Considering the real-world practice, increasing neoadjuvant chemotherapy in HER2 and TNBC patients, this is one of most important limitation of our study, as we only included patients received upfront surgery. Further study using other confirmatory method than IHC would be required the YAP1 activation in various subtype of breast cancer with pre-treatment biopsy sample.

             In this study, we focused on the nuclear localization of YAP1 and the matrix stiffness. As breast cancer appeared to be mostly appropriate samples since we had the surgical tumor tissue and preoperative elastography data that reflect the tumor stiffness. Although we showed the correlation of clinical parameter and YAP1 IHC, further in vitro study with precise mechanical stress and the measurement of nuclear YAP1 might support our result.

Reviewer 2 Report

The authors have presented a comprehensive study to investigate the correlation between nuclear expression of YAP1 and tumor stiffness. The manuscript is overall well written. I have the following suggestions to improve the discussion.

In discussion section the authors should include the correlation status between YAP1 and molecular subtype/ HR status/ TIL level/ Stroma type and associate with tumor stiffness. In this way the preclinical scientist can prepare a hypothesis on the probable mechanism to induce tumor stiffness.

The authors should highlight if there is relation between tumor stiffness-YAP1 expression is associated with immunosuppression.

It may be important to add the therapeutic regime of the patients in the supplementary files.

It would be great if the authors can briefly discuss on the dpendence of tuor stiffness on YAP1 expression or vice versa (YAP1 is a marker for tumor stiffness in breast cancer).

Table 3,4 and 5 can be moved to the supplementary material. 

Author Response

Reviewer 2

The authors have presented a comprehensive study to investigate the correlation between nuclear expression of YAP1 and tumor stiffness. The manuscript is overall well written. I have the following suggestions to improve the discussion.

In discussion section the authors should include the correlation status between YAP1 and molecular subtype/ HR status/ TIL level/ Stroma type and associate with tumor stiffness. In this way the preclinical scientist can prepare a hypothesis on the probable mechanism to induce tumor stiffness.

[Answer]

Thank you for your kind and valuable comments. We further described the correlation status of variable parameters with tumor stiffness in the discussion section (lines 321-322)

The authors should highlight if there is relation between tumor stiffness-YAP1 expression is associated with immunosuppression.

[Answer]

Thank you for your kind and valuable comments that improve our manuscript. We further check the other studies regarding tumor stiffness-YAP1 expression and immunosuppression, and mentioned their relationship in the discussion section as below: (lines 307-315)

Contribution of YAP1 in tumor immunity has not been clearly established. However, several studies investigated that YAP1 expression is associated with increased immune cell infiltration in various solid tumors including pancreatic cancer [1], ovarian cancer [2], and HCC [3] etc. So far, direct correlation with matrix stiffness – YAP1- immunosuppression has not been well studied. Matrix stiffness itself contribute to immune evasion of tumor cells by reduce cellularity and function of cytotoxic T-cells as well as enhance immunosuppressive macrophage polarization [4]. Recent in vitro study showed that mechanical stiffness promotes dendritic cells and elicit adaptive immune response in tumor immunity via TAZ, which is YAP homologous [5].

  1. Murakami, S.; Shahbazian, D.; Surana, R.; Zhang, W.; Chen, H.; Graham, G.T.; White, S.M.; Weiner, L.M.; Yi, C. Yes-associated protein mediates immune reprogramming in pancreatic ductal adenocarcinoma. Oncogene 2017, 36, 1232-1244.
  2. Sarkar, S.; Bristow, C.A.; Dey, P.; Rai, K.; Perets, R.; Ramirez-Cardenas, A.; Malasi, S.; Huang-Hobbs, E.; Haemmerle, M.; Wu, S.Y. et al. Prkci promotes immune suppression in ovarian cancer. Genes Dev 2017, 31, 1109-1121.
  3. Guo, X.; Zhao, Y.; Yan, H.; Yang, Y.; Shen, S.; Dai, X.; Ji, X.; Ji, F.; Gong, X.G.; Li, L. et al. Single tumor-initiating cells evade immune clearance by recruiting type ii macrophages. Genes Dev 2017, 31, 247-259.
  4. Jiang, Y.; Zhang, H.; Wang, J.; Liu, Y.; Luo, T.; Hua, H. Targeting extracellular matrix stiffness and mechanotransducers to improve cancer therapy. J Hematol Oncol 2022, 15, 34.
  5. Chakraborty, M.; Chu, K.; Shrestha, A.; Revelo, X.S.; Zhang, X.; Gold, M.J.; Khan, S.; Lee, M.; Huang, C.; Akbari, M. et al. Mechanical stiffness controls dendritic cell metabolism and function. Cell Rep 2021, 34, 108609.

It may be important to add the therapeutic regime of the patients in the supplementary files.

[Answer]

Thank you for your suggestion. We added the information regarding therapeutic regimens of the patients as supplementary file (supplementary table 1).

It would be great if the authors can briefly discuss on the dependence of tumor stiffness on YAP1 expression or vice versa (YAP1 is a marker for tumor stiffness in breast cancer).

[Answer]

Thank you for your valuable comments that improve our manuscript. We added a brief comment the possibility of surrogate role of YAP1 for tumor stiffness. (lines 274-275)

Table 3,4 and 5 can be moved to the supplementary material. 

[Answer]

Thank you for your comments. We moved the tables to the supplementary material. (supplementary table 2, 3, and 4)

Reviewer 3 Report

In the article entitled, ‘Correlation of yes-associated protein 1 with stroma type and tumor stiffness in hormone-receptor positive breast cancer’ by Lee et al., the authors have assessed the correlation between tumor tissue stiffness and YAP1 in HR+ Breast cancer tissues. The authors have further studied the association between tumor stiffness and other parameters such as tumor-infiltrating lymphocytes and tumor-stromal ratio. Overall, the study has been performed well and the findings are interesting. However, there are a few concerns that the authors must address before publication in the journal.

Major concerns:  

1. The authors can assess patients treated with chemotherapeutic agents that suppress/deplete the stroma and hence the tumor stiffness and evaluate the effect on YAP expression.

2. The authors must check the activation of YAP signaling in at least a few patient samples if available to validate the findings. 

Minor concerns:

1. The authors have not included many references which are relevant to the study. These must be included (PMID: 31015465, 31785406).

Author Response

Reviewer 3

In the article entitled, ‘Correlation of yes-associated protein 1 with stroma type and tumor stiffness in hormone-receptor positive breast cancer’ by Lee et al., the authors have assessed the correlation between tumor tissue stiffness and YAP1 in HR+ Breast cancer tissues. The authors have further studied the association between tumor stiffness and other parameters such as tumor-infiltrating lymphocytes and tumor-stromal ratio. Overall, the study has been performed well and the findings are interesting. However, there are a few concerns that the authors must address before publication in the journal.

Major concerns:  

  1. The authors can assess patients treated with chemotherapeutic agents that suppress/deplete the stroma and hence the tumor stiffness and evaluate the effect on YAP expression.

[Answer]

Thank you for your valuable comments. Stroma remodeling upon the chemotherapy and the effect on YAP1 expression would be worth to study. However, in this study, only patients received upfront surgery were included because we wanted to evaluate the nuclear YAP1 expression in the whole surgical specimens.

Chemotherapeutic agents could deplete stroma as well as tumor cells. In those cases, evaluation of YAP1 expression in the residual tumor cells might be difficult. Rather, aside the YAP1 expression, investigation of pre-operative tumor stiffness in neoadjuvant chemotherapy-treat tumor and the treatment response seems to be interesting to investigate, and might be the further subject of our team.

  1. The authors must check the activation of YAP signaling in at least a few patient samples if available to validate the findings. 

[Answer]

Thank you for your valuable comments that improve our manuscript.

Further check on the activated YAP1 and downstream signaling may lead more solid conclusion to our study. However, it appears to be a difficult validation of YAP1 signaling effect accurately. Nuclear localization of YAP1 is known to activate multiple cancer-associated genes, and YAP1 activation with poor prognosis in various solid tumors is considered to be a result derived from all the complex effects of cancer-associated gene that put together. Moreover, as downstream cancer-associated genes are variable, selection bias might be inevitable.

We agree that one of major limitation is lack of clinical impact of YAP1 expression in breast cancer. As HR+ breast cancer patients have long term survival in general, no significant prognostic events has been occurred in this study. However, we further plan to analyze the YAP1 expression and its correlation with oncotype Dx score in HR+ breast cancer, since oncotype Dx score could stratify the patients into prognostic risk groups.

Minor concerns:

  1. The authors have not included many references which are relevant to the study. These must be included (PMID: 31015465, 31785406).

[Answer]

Thank you very much for your kind comment and careful suggestion that improve our manuscript. We also read the article of Joanna Y. Lee et al. (PMID 31015465). In that study, authors found that YAP1 is inactivated in the pre-invasive lesion, and also, YAP1 and mechanotransduction had no relationship in breast cancer in 3D culture under no stress fibers. The previous study was in vitro study without TME like stromal cells or TILs, and only mechanical pressure was put on the cultured cancer cells. This might lead the different result with in vivo experience. In our study, we found that the correlation between YAP1 and stromal stiffness was well delineated with non-collagenous stroma, which means cellular components of stroma might play some roles in YAP1 activation as well as in stiffen the matrix.

Another reference, regarding drug resistance (PMID 31785406) is also very important in breast cancer. In that study, drug resistance of breast cancer cells was increased along the ECM stiffness. Authors showed that stiffness induced YAP1 nuclear translocation, that mediates drug resistance gene expression, and also suggested that inhibition of YAP1 could regulate drug resistance of breast cancer cells.

We included both valuable references in our manuscript and briefly described in the discussion section. (lines 302-306 and lines 360-364)

Reviewer 4 Report

key word should be in alphabetic order 

The information in the section "Simple summary" also appears at the beginning of the paper. There is no need to be repeated

Author Response

Reviewer 4

key word should be in alphabetic order 

The information in the section "Simple summary" also appears at the beginning of the paper. There is no need to be repeated

[Answer]

Thank you for your kind comments that improve our manuscript.

Key words have been rearranged in alphabetical order. (lines 38-39)

Repeated part of simple summary was removed and revised. (line 16)

Round 2

Reviewer 1 Report

All the points I raised were addressed by the authors.

Author Response

Thank you for your kind comment.

We further mentioned the limited sample size in the discussion section (lines 376-377).

Reviewer 3 Report

The authors have answered the concerns raised. The manuscript may be accepted in the present format.

Author Response

Thank you for your kind comment.

We further mentioned the indirect assessment of YAP1 acitation in the discussion section (lines 355-361) .